# A Genotype-Phenotype Study of High-Resolution *FMR1* Nucleic Acid and Protein Analyses in Fragile X Patients with Neurobehavioral Assessments

**DOI:** 10.3390/brainsci10100694

**Published:** 2020-09-30

**Authors:** Dejan B. Budimirovic, Annette Schlageter, Stela Filipovic-Sadic, Dragana D. Protic, Eran Bram, E. Mark Mahone, Kimberly Nicholson, Kristen Culp, Kamyab Javanmardi, Jon Kemppainen, Andrew Hadd, Kevin Sharp, Tatyana Adayev, Giuseppe LaFauci, Carl Dobkin, Lili Zhou, William Ted Brown, Elizabeth Berry-Kravis, Walter E. Kaufmann, Gary J. Latham

**Affiliations:** 1Departments of Psychiatry and Neurogenetics, Fragile X Clinic, Kennedy Krieger Institute, Baltimore, MD 21205, USA; dragana.protic@med.bg.ac.rs; 2Department of Psychiatry & Behavioral Sciences-Child Psychiatry, Johns Hopkins School of Medicine, Baltimore, MD 21205, USA; 3Asuragen, Inc., Austin, TX 78744, USA; sydnette1@yahoo.com (A.S.); sfilipovicsadic@asuragen.com (S.F.-S.); eranbram@gmail.com (E.B.); knicholson@asuragen.com (K.N.); kculp@asuragen.com (K.C.); mohammadjca1@gmail.com (K.J.); jkemppainen@asuragen.com (J.K.); ahadd@natera.com (A.H.); 4School of Medicine, University of Belgrade, 11000 Belgrade, Serbia; 5Department of Neuropsychology, Kennedy Krieger Institute, Baltimore, MD 21205, USA; mmahone83@gmail.com; 6Department of Pediatrics, Rush University Medical Center, Chicago, IL 60612, USA; kevin_j_sharp@rush.edu (K.S.); lili_zhou@rush.edu (L.Z.); elizabeth_berry-kravis@rush.edu (E.B.-K.); 7Institute for Basic Research in Developmental Disabilities, Staten Island, NY 10314, USA; tatyana.adayev@opwdd.ny.gov (T.A.); giuseppe.x.lafauci@opwdd.ny.gov (G.L.); carl.dobkin@omr.state.ny.us (C.D.); wtbibr@aol.com (W.T.B.); 8Departments of Pediatrics, Neurological Sciences and Biochemistry, Rush University Medical Center, Chicago, IL 60612, USA; 9Department of Human Genetics, Emory University School of Medicine, Atlanta, GA 30322, USA; walter.e.kaufmann@emory.edu; 10Boston Children’s Hospital, Harvard Medical School, Boston, MA 02115, USA

**Keywords:** autism spectrum disorder, fragile X syndrome, FMRP, *FMR1*, PCR

## Abstract

Fragile X syndrome (FXS) is caused by silencing of the *FMR1* gene, which encodes a protein with a critical role in synaptic plasticity. The molecular abnormality underlying *FMR1* silencing, CGG repeat expansion, is well characterized; however, delineation of the pathway from DNA to RNA to protein using biosamples from well characterized patients with FXS is limited. Since FXS is a common and prototypical genetic disorder associated with intellectual disability (ID) and autism spectrum disorder (ASD), a comprehensive assessment of the *FMR1* DNA-RNA-protein pathway and its correlations with the neurobehavioral phenotype is a priority. We applied nine sensitive and quantitative assays evaluating *FMR1* DNA, RNA, and FMRP parameters to a reference set of cell lines representing the range of *FMR1* expansions. We then used the most informative of these assays on blood and buccal specimens from cohorts of patients with different *FMR1* expansions, with emphasis on those with FXS (N = 42 total, N = 31 with FMRP measurements). The group with FMRP data was also evaluated comprehensively in terms of its neurobehavioral profile, which allowed molecular–neurobehavioral correlations. *FMR1* CGG repeat expansions, methylation levels, and FMRP levels, in both cell lines and blood samples, were consistent with findings of previous *FMR1* genomic and protein studies. They also demonstrated a high level of agreement between blood and buccal specimens. These assays further corroborated previous reports of the relatively high prevalence of methylation mosaicism (slightly over 50% of the samples). Molecular-neurobehavioral correlations confirmed the inverse relationship between overall severity of the FXS phenotype and decrease in FMRP levels (N = 26 males, mean 4.2 ± 3.3 pg FMRP/ng genomic DNA). Other intriguing findings included a significant relationship between the diagnosis of FXS with ASD and two-fold lower levels of FMRP (mean 2.8 ± 1.3 pg FMRP/ng genomic DNA, *p* = 0.04), in particular observed in younger age- and IQ-adjusted males (mean age 6.9 ± 0.9 years with mean 3.2 ± 1.2 pg FMRP/ng genomic DNA, 57% with severe ASD), compared to FXS without ASD. Those with severe ID had even lower FMRP levels independent of ASD status in the male-only subset. The results underscore the link between *FMR1* expansion, gene methylation, and FMRP deficit. The association between FMRP deficiency and overall severity of the neurobehavioral phenotype invites follow up studies in larger patient cohorts. They would be valuable to confirm and potentially extend our initial findings of the relationship between ASD and other neurobehavioral features and the magnitude of FMRP deficit. Molecular profiling of individuals with FXS may have important implications in research and clinical practice.

## 1. Introduction

Fragile X syndrome (FXS) is caused by a full mutation (FM, >200 CGGs) expansion in the promoter region of the *FMR1* (fragile X mental retardation 1) gene. The expansion of the mutation leads to atypical methylation, transcriptional silencing, and ultimately deficiency of the *FMR1*-encoded protein (fragile X mental retardation protein, FMRP) [1]. The degree of methylation is influenced by the number of tandem CGG trinucleotide repeats in the 5’ UTR. Normal genotypes (up to 44 CGGs) and the majority of premutation genotypes (PM, 55-200 CGGs) are associated with an unmethylated or partially methylated gene that expresses *FMR1* transcripts [2] and its protein (FMRP) in the brain and other tissues [3]. In contrast, in individuals with the FM genotype, *FMR1* is typically fully or mostly methylated, while transcript and protein expression are markedly reduced. FMRP interacts with about 4% of total mammalian brain mRNAs [2,4,5] and regulates many proteins involved in synaptic development and function [6]. As an X-linked disorder, females with FXS often have a less severe phenotype than males since FMRP may be expressed from approximately 50% of the cells due to random X-inactivation [7]. Levels of FMRP in the blood of patients with FXS have been positively correlated with cognitive performance, specifically intelligence quotient (IQ) and adaptive behavior [8,9,10].

*FMR1* gene expansions are complex, exhibiting size and methylation variability (i.e., mosaicism); skewed X-inactivation may add on another layer of complexity in females. These factors may affect the level of FMRP deficiency and phenotype severity [7,11,12,13]. Both size and methylation mosaicism (MM) can vary in different tissues within an individual [14,15,16], which further complicates the interpretation of phenotypical heterogeneity. For example, previous studies have highlighted differences in the *FMR1* genotypes between blood cells (of mesodermal origin) and fibroblast or buccal epithelial cells (of ectodermal origin) from the same individual [14,17]. As FXS mainly affects the brain, these discrepancies are highly relevant: neurons and other cells have an ectodermal origin whereas *FMR1* and FMRP assays are generally based on blood samples.

*FMR1* genomic profiles are complex and merit characterization in FXS. In particular, FMRP expression in the brain is the ultimate factor determining the severity of the neurobehavioral phenotype [18]. Intriguingly, it is estimated that 10–20% of normal FMRP expression is sufficient for a cognitive performance at the borderline IQ level [19]. Studies applying sensitive methods have revealed that many individuals with FM alleles are size and/or methylation mosaics. These individuals retain the capability of producing at least some FMRP as they are harboring both differentially methylated PM alleles and/or FM alleles [20,21,22].

In addition to genotypical and molecular phenotypical (i.e., FMRP) complexity, FXS is heterogeneous in its physical and neurobehavioral manifestations [23]. This is particularly evident in the range of neurobehavioral abnormalities associated with FM, such as the most prevalent anxiety, along with hyperarousal, impulsivity and other attentional network deficits, aggression, and self-injurious behavior often accompanied by irritability [24]. FXS is also frequently comorbid with autism spectrum disorder (ASD; 50% of male and 20% of female) [25,26,27]. Indeed, evidence supports a partial overlap between the pathogenetic mechanisms that lead to FXS and ASD in the general population [28,29]. Additionally, many proteins that interact with FMRP are also associated with idiopathic ASD [30,31,32]. Lower FMRP levels have also been documented in some research samples of individuals with FXS and ASD (vs. FXS only) and idiopathic ASD [33]; thus, FMRP deficits may impact molecular pathways and synaptic processes [6] underlying some aspects of autistic behavior [29]. Importantly, greater severity and lower functioning is associated with ASD co-morbidity in FXS [26,34]. A higher prevalence of seizures, sleep problems, and co-occurring problem behaviors, especially aggressive/disruptive behavior, is found in the pediatric population with FXS and ASD than in the FXS only population [35]. Nonetheless, it is difficult to determine the selective contribution to ASD as most of these individuals with FXS also show lower cognitive performance [32]. The aforementioned heterogeneity and the difficulties in interpreting blood-based *FMR1* and FMRP profiles in FXS is quite compelling, further underscoring the need to improve molecular assays and molecular-phenotypical analyses.

Namely, integration of diagnostic genomic data with complementary *FMR1* assays and more accurate FMRP profiles is necessary [36]. The integration of these data can clarify the relationships between genotype and protein expression and the neurobehavioral phenotype [37]. Such links may improve disease prognostics and characterization of response to pharmacological and other therapeutic interventions [11,36], as well as help stratify patients with FXS in clinical trials [38]. Recently, a series of *FMR1*-based molecular methods quantifying genetic and epigenetic features of the mutation with greater sensitivity, specificity, and breadth have been reported [20,21,39,40,41]. These *FMR1* assays can be applied across different specimen types, such as blood and buccal epithelium, and combined with FMRP measurements [42,43]. Specifically, advanced PCR-based methods can reliably amplify small and large *FMR1* repeat expansions, detect low-level repeat size mosaicism, determine the number and the sequence context of sequences that interrupt the repeat tract, quantify X-inactivation, and measure skewed *FMR1* gene silencing that was masked using less sensitive techniques such as Southern blotting [11,20,21,39,44,45,46]. In addition, more precise FMRP assays allow a better delineation of the range of *FMR1* protein levels and, consequently, more accurate *FMR1*–FMRP correlations [42,43,47]. This integrated approach was recently used to explore repeat instability in a reprogrammed stem cell model of patients with fragile X-associated primary ovarian insufficiency (FXPOI) [42].

In the present investigation, we report results from a comprehensive evaluation of *FMR1* and FMRP profiles and their associations with the FXS neurobehavioral phenotype. We compared nine complementary assays for assessing *FMR1* DNA, RNA, and protein, using a reference set of well-characterized *FMR1* cell lines. We then applied data from the most informative assays to examine present molecular–neurobehavioral relationships in an independent cohort of patients with FXS with and without ASD. In addition to these primary analyses, we also contrasted molecular profiles in matched blood and buccal specimens in order to include their value in informing FXS phenotypical variability.

## 2. Materials and Methods

### 2.1. Participants

Two cohorts were included in the molecular and molecular-phenotypical analyses. The first was a Reference Cohort of 11 individuals, with and without FXS, that provided whole blood specimens at Rush University Medical Center (RUMC) under approved Institutional Review Board (IRB) protocols. These five males and six females represented a range of *FMR1* genotypes, from normal to FM; their samples were used to generate genomic DNA (gDNA) for different genomic studies (repeat-primed PCR (RP-PCR), methylation PCR (mPCR), AGG interruption analysis, and Southern blot analysis) and to create lymphoblastoid cell lines and other durable control material for evaluating, comparing, and contrasting *FMR1* DNA and protein analyses.

The second or Clinical Cohort consisted of 42 patients with a range of *FMR1* expansions (Table 1), who were randomly recruited from a pool of patients assessed at the Kennedy Krieger Institute (KKI)’s Fragile X Clinic between 2013–2016. Table 1 presents an overview of the Clinical Cohort, including its demographic features. Detailed genomic profiling was performed for the subset of patients with FM (FM–ALL Subcohort, N = 37; Table 2); most of these participants also had FMRP measurements (FM–FMRP Subcohort, N = 31; Table 1), detailed neurobehavioral assessments, and other genomic assays (Table 2 and Table 3). All protocols were approved by the Johns Hopkins Medicine’s IRB.

Age distribution was comparable among individuals in the Clinical Cohort. The majority of the participants were Caucasian males (% males/% Caucasian): 78/76 for the entire Clinical Cohort, 84/74 for the FM–ALL Subcohort, and 84/77 for the FM–FMRP Subcohort. As shown in Table 1, there was a small subgroup of individuals with PM that consisted of two males and three females.

### 2.2. Materials

#### 2.2.1. *FMR1* Reference Cell Lines

The Reference Cohort consisted of three individuals in the normal *FMR1* CGG range, five with PM, and three with FM, each of whom contributed a lymphoblastoid cell line. These lines were established at RUMC using standard techniques (i.e., peripheral blood mononuclear cell Epstein–Barr virus transformation). Transformed cell lines were expanded, cryopreserved, and provided as frozen stocks to the Asuragen team (Austin, TX, USA) for further expansion. Cell lines were grown in suspension at 37 °C in RPMI 1640 (Gibco, Thermo Fisher Scientific, Waltham, MA, USA) supplemented with 15% fetal bovine serum, 2 mM L-glutamine, and penicillin and streptomycin. Viability and total cell counts were determined using a standard trypan blue stain procedure and a phase hemacytometer (VWR, Radnor, PA, USA); bacterial contamination was assessed using a Universal Mycoplasma Detection Kit (ATCC, Manassas, VA, USA). After being grown to confluence (~8 × 10^5^ cells/mL), cells were harvested for DNA, RNA, and total protein isolation.

#### 2.2.2. Whole Blood Collection and Preservation

Whole blood was collected using: (i) EDTA vacutainer tubes, as the whole liquid blood for initial 20 subjects of the Clinical Cohort; (ii) FTA cards, preserved blood saturated spot cards for all (N = 42) subjects (GE Healthcare Life Sciences, Marlborough, MA, USA); and/or (iii) 903 Specimen Collection Paper/Whatman protein saver cards for a total of 36/42 subjects (GE Healthcare Life Sciences, Marlborough, MA, USA). Both the whole liquid blood and FTA cards were analyzed for the first 20/42 subjects. This approach demonstrated that *FMR1* CGG repeat quantification and DNA methylation analysis were not affected by the method of the blood collection or storage; thus, the FTA cards were selected as a preferred sample type for the remaining 22/42 subjects in the study. FMRP analysis, however, was not supported by specimens preserved on FTA cards, which prompted the use of the 903 protein saver cards as an additional specimen collection. Cards were spotted with ~125 µL of whole blood (EDTA tubes) and stored at room temperature. Specimens from each subject were shipped to Asuragen for subsequent molecular and protein testing (*FMR1* and FMRP analysis).

#### 2.2.3. Matched Blood and Buccal Specimens

Matched whole blood and buccal cell specimens were collected from 42 patients from the Clinical Cohort. Buccal cell samples were collected using ORAcollect·DNA (OCR-100; DNA Genotek, Ottawa, CA) following the manufacturer’s instructions. Blood samples were obtained using standard clinical procedures. All samples were obtained following informed consent and according to protocols approved by the Johns Hopkins Medicine’s IRB.

#### 2.2.4. Genomic DNA Isolation and Primary Characterization

Genomic DNA was isolated from the whole blood specimens (Clinical Cohort, N = 42), FTA cards, buccal swabs, and cell lines described above using the DNeasy Blood and Tissue Kit (Qiagen, Germantown, MD, USA) in accordance with the manufacturer’s instructions. The concentration of gDNA was determined using spectrophotometry (NanoDrop, Thermo Fisher Scientific, Waltham, MA, USA). The level of intact or non-fragmented DNA was assessed by agarose gel electrophoresis (AGE, E-Gels Precast Agarose Gels; Thermo Fisher Scientific, Waltham, MA, USA).

To isolate multiple analytes (DNA, RNA) simultaneously from the cell lines, we used the AllPrep DNA/RNA Mini Kit (Qiagen, Germantown, MD, USA) following the manufacturer’s instructions. Nucleic acid concentration was determined by spectrophotometry (NanoDrop, Thermo Fisher Scientific, Waltham, MA, USA). Additionally, RNA integrity was determined on the 2100 Bioanalyzer (Agilent, Santa Clara, CA, USA) using standard procedures. The minimum RNA Integrity Number (RIN) for all samples was 9.6.

### 2.3. Molecular Assessments

Table 2 depicts samples and analyses performed with both the Reference Cohort and Clinical Cohort.

#### 2.3.1. CGG Repeat Genotyping and AGG Interruption Analysis

*FMR1* genotypes from cell-line and blood sample gDNA were determined using triplet repeat-primed PCR, followed by fragment analysis of amplicons by capillary electrophoresis (CE) on an ABI 3500xl Genetic Analyzer (Thermo Fisher Scientific, Waltham, MA, USA), and Sanger sequencing. PCR-based CGG repeat genotyping was performed with both two-primer and three-primer *FMR1* PCR/CE ((*FMR1* gene-specific PCR/CE (GS-PCR/CE) and repeat-primed PCR/CE (RP-PCR/CE), AmplideX^®^ PCR/CE *FMR1* Kit (Asuragen, Austin, TX, USA)) [20,21]. Cell-line gDNA samples were also analyzed using two-primer *FMR1* PCR with products resolved by agarose gel electrophoresis (GS-PCR/AGE) [21]; this permitted sizing of repeat expansions larger than ~200 CGG that cannot be sized within the resolution limits of CE. Samples were annotated as size mosaics if they manifested peaks in two different size categories such as PM and FM with a peak signal intensity greater than 75 relative fluorescence units (RFU) by CE. The number and sequence context of interrupting AGG sequences in the repeat tract was determined using Xpansion Interpreter^®^, a PCR-based method (Asuragen, Austin, TX, USA) [40,41,45,48].

#### 2.3.2. Methylation PCR Analysis

The AmplideX mPCR *FMR1* Kit (Asuragen, Austin, TX, USA) was used to quantify relative allele-specific DNA methylation of the *FMR1* gene [39,49] on gDNA from blood and buccal samples or cell lines. Both X-inactivation and methylation of the expanded allele could be assessed using this method, revealing completely or partially methylated states [44]. A sample was defined as a methylation mosaic (MM) if it contained FM or PM fragment peak(s) above 75 RFU by CE that was/were <80% methylated in at least one sample type (buccal swab or whole blood).

#### 2.3.3. Southern Blot Analysis

Southern blot analysis was performed utilizing gDNA isolated either from the whole blood samples or from the cell lines. DNA was digested with *Eco*RI and *NruI* and separated on an agarose gel. After the DNA transfer, the membranes were hybridized with the *FMR1*-specific StB12.3 genomic probe and imaged according to published procedures [50].

#### 2.3.4. *FMR1* mRNA Analysis

*FMR1* transcript expression was evaluated via quantitative real-time PCR (qPCR) for all cell line samples. A total of 2.5 µg of purified total mRNA was converted to cDNA using a reverse transcription (RT) protocol. The RT reaction was incubated at 42 °C for 45 min followed by 10 min at 93 °C, and then rapidly cooled to 4 °C. *FMR1* mRNA specific primers (Integrated DNA Technologies, Coralville, IA, USA) (Forward Primer: 5′-TATGCAGCATGTGATGCAACT-3′, Reverse Primer: 5′-TTGTGGCAGGTTTGTTGGGAT-3′) for use with the KAPA SYBR Fast qPCR kit (KAPA Biosystems, Wilmington, MA, USA) were applied according to the manufacturer’s instructions. Each sample was run at two input concentrations of 31 ng/µL and 6.25 ng/µL, respectively. *FMR1* mRNA expression was quantified relative to expression of the control housekeeping gene *USP33* [42].

#### 2.3.5. Quantitative FMRP Analysis

FMRP levels were determined using a quantitative FMRP (qFMRP) assay developed by LaFauci et al. [43]. This assay utilizes a recombinant FMRP peptide (GST-SR7) as the standard for quantification of FMRP level. Briefly, for the protein extraction, eight 3 mm punches from blood spotted and dried on Whatman 903 paper were added to 200 µL M-PER solution (Thermo Fisher Scientific, Waltham, MA, USA) supplemented with protease inhibitors (cOmplete, Mini, EDTA-free; Roche Applied Science, Indianapolis, IN, USA). After incubation for 3 h at room temperature, samples were briefly centrifuged at 10,000× *g* to remove cell debris. Two 50 µL aliquots of the eluate were independently analyzed in the qFMRP assay using liquid bead array (Luminex, Austin, TX, USA) following the method described by Gustin et al. [42]. The FMRP concentration in each sample was determined after comparing the relative fluorescent intensity against an 11-point standard curve constructed using 0.28 to 280 pM of the GST-SR7 peptide, along with a blank. To normalize FMRP concentration across blood specimens, gDNA was used as a proxy for cell count. For each sample, gDNA was quantified using the PicoGreen Quant-iT™ HS kit (Thermo Fisher Scientific, Waltham, MA, USA) and the FMRP level was standardized to the gDNA amount. Where relevant, the total protein concentration of the lysates was determined using a Pierce BCA Protein Assay Kit (Thermo Fisher Scientific, Waltham, MA, USA).

### 2.4. Clinical Assessments

Table 3 summarizes the neurobehavioral assessments carried out on the FM–FMRP cohort (N = 31). They included (i) presence (diagnosis) of ASD, social anxiety (SA), and/or unspecified anxiety, (ii) level of intellectual functioning/intellectual disability (ID), as determined by a Full Scale Intelligence Quotient (FSIQ), (iii) level of adaptive functioning, (iv) severity of problem behaviors, (v) overall clinical severity, as determined by the Clinical Global Impression-Severity scale (CGI-S), and (vi) use of antipsychotics.

The following assessment tools were employed:(i)Diagnostic and Statistical Manual-5th Edition criteria (DSM-5) [51], supplemented by Autism Diagnostic Observation Schedule (ADOS) assessments available in males with FM, were used to diagnose ASD. All males with FM in the study diagnosed with ASD had both DSM-5 and ADOS assessments, respectively. The diagnosis of ASD (and Non-ASD) was made clinically, and confirmed longitudinally, for all subjects in the Clinical Cohort by a clinician (DBB) with expertise in idiopathic ASD, and ASD in FXS [25,26,27,32,34]. Diagnoses of SA and unspecified anxiety were also made using DSM-5 criteria [27,32]. SA include a substantial social inhibition (shyness) accompanied by a broad range of fear of negative evaluation by others, which may be embarrassing, lead to rejection or offend others such as the expression of anger toward others. The “Fragile-X handshake” and various forms of “escape” behaviors in familiar or particularly unfamiliar situations are common as well. (See section (v) for profiling of severity/level of SA diagnosis).(ii)A FSIQ was determined by the Stanford–Binet Intelligence Scales-5th Edition (SB-5) for 15/31 individuals, the Wechsler Preschool and Primary Scale of Intelligence (WPPSI) for 1/31, the Wechsler Intelligence Scale for Children (Fourth and Fifth Edition, WISC-IV and WISC-V) for 7/31, The Differential Ability Scales (DAS) for 4/31 and the Mullen Scales of Early Learning for 4/31 [38]. To address the skewed effect of FSIQ-standard or other score testing, raw-score based z-score calculations from the IQ subtests were used by a senior neuropsychologist (EMM) to generate extended FSIQ values [52]. When there were several administrations of a test, the most current one was used for data analysis. Alternatively, if the scores were highly discrepant, estimation was made via interpolation of the two scores. To determine level of ID, the extended FSIQ scores were also used instead of adaptive skill scores because the former, as scaled measures, better reflect the range of cognitive abilities. Subjects were assigned to one of four ID levels: normal range (FSIQ score ≥ 70), mild ID (FSIQ: 55–69), moderate (FSIQ: 35–54), and severe ID (FSIQ < 35).(iii)Adaptive functioning was assessed by using adaptive skill scales, which included the Vineland Adaptive Behavior Scales-Second Edition (VABS–II) for most participants, the Adaptive Behavior Assessment System Second and Third Editions (ABAS-2 and -3) for 10 individuals, and Scales of Independent Behavior-Revised (SIB-R) for 2 participants.(iv)Problem behaviors were assessed by the Aberrant Behavior Checklist-Community Edition (ABC-C) adapted for FXS (ABC-CFX), which applies a subscale scoring algorithm developed specifically for the disorder and yields six subscales [53]: (i) Irritability, (ii) Lethargy/Social Withdrawal, (iii) Stereotypic Behavior, (iv) Hyperactivity, (v) Inappropriate Speech, and (vi) Social Avoidance [53,54]. The ABC-CFX has been applied as a primary outcome measure in multiple observational and interventional studies in FXS (reviewed in [38]).(v)The CGI-S score evaluates the overall impairment of a patient using the clinician’s past experience with patients who have the same diagnosis as a reference. Possible ratings of the CGI-S are as follows: 1—normal, not at all ill, 2—borderline ill, 3—mildly ill, 4—moderately ill, 5—markedly ill, 6—severely ill, and 7—extremely ill. The CGI approach was also applied separately to profile the severity/level of SA (CGI-SANX); based on CGI-SANX scores, two categories were defined: ≥5 (severe) and ≤4 (mild-moderate).(vi)A patient’s use of antipsychotics was determined through health records and marked as “there is” or “there is not” (yes/no) use of this class of drugs.

### 2.5. Statistical Analysis

Statistical analysis was performed using IBM SPSS Statistics version 25 (IBM Corporation, Armonk, NY, USA) or JMP version 14 (SAS Institute, Cary, NC, USA). Descriptive statistics included frequency (percent) of nominal variables and median, mean, standard deviation (SD), and range for continuous variables. Tests of normality and homogeneity of variances were also performed. Depending on data distribution, either parametric or non-parametric tests were applied. The Chi square test was used to test differences between nominal variables (frequencies). Pearson’s correlation coefficient was used as a measurement of the strength of the linear relationship between normally distributed variables. Welch’s *t*-test for unequal variances and equivalent non-parametric tests (Mann–Whitney) were performed to compare means of two samples. Welch’s *t*-test maintains type I error rates close to nominal for unequal variances and for unequal sample sizes under normality. Significance was indicated by *p* ≤ 0.05 and high significance by *p* ≤ 0.01.

## 3. Results

### 3.1. FMR1 DNA, RNA, and Protein Assays Assessed in a Reference Cell-Line Cohort

High-resolution, sensitive, and specific analyses of the gene and its protein are necessary to understand genotype–phenotype links and their potential clinical impact on FXS. We first compared nine *FMR1* DNA, RNA, or protein-based assays (Table 2) using 11 well-characterized lymphoblastoid cell lines (Table 4) from patients with FXS. These nine assays included five *FMR1* PCR assays [20,21,39,40,41,42,43,44,45,46] (Figure 1). As shown in Figure 1A,B, the results from both GS-PCR/AGE and RP-PCR/CE were in agreement, both for genotype category (i.e., normal, PM, or FM) and number of CGG repeats (within the known sizing resolution of AGE and CE). Sanger sequencing of selected expanded samples (Figure 1: RU06 and RU08) confirmed those findings. Analysis of interrupting AGG elements were consistent with previous reports demonstrating these interspersions were more highly represented in unexpanded alleles [40,41,55]. We note that the primary CGG sizing and AGG interruption genotypes in the immortalized cell lines were preserved when compared to results using these same assays for the original patient blood samples.

Next, *FMR1* mPCR was used to determine the extent of gene methylation and silencing. As expected, the normal and PM cell-lines were unmethylated (males) or partially methylated (females). Two of the three FM cell lines were nearly completely methylated consistent with inactivation of *FMR1* in FXS. However, one FM cell line (RU08) was unmethylated (Figure 1C). Southern blot analysis of the original whole blood sample also showed lack of methylation (Table 4). To investigate this finding further, we isolated RNA and quantified the primary *FMR1* mRNA isoform on all 11 cell lines. Since a lack of *FMR1* methylation would be expected to be associated with transcription of the gene, we anticipated that *FMR1* RT-qPCR would demonstrate substantially greater transcript levels for RU08 compared to the other two FM cell-lines with methylated *FMR1*. Indeed, this was the case (Figure 1D). The level of *FMR1* expression was also consistent with methylation status for the other cell-line samples (Figure 1D). We also compared methylation levels of FM cell lines with the corresponding blood samples used for cell immortalization and found nearly identical percent methylation for all three sample pairs.

Finally, we measured FMRP levels for each cell line to assess the impact of repeat expansions and other *FMR1* DNA and RNA parameters on translation. Lysates of the 11 cell lines were analyzed by a previously established antibody-based qFMRP liquid bead array assay [42,43]. As shown in Figure 1E, FMRP levels were found to be well above the limit of detection for nine of the 11 cell lines including the unmethylated FM cell line (RU08). The two FM samples (RU03 and RU10) with pronounced gene methylation showed absence of FMRP. Thus, the results indicate that this set of multianalyte assays can quantify *FMR1* gene and FMRP parameters across the range of *FMR1* categorical genotypes with complementary results. Further, the molecular profiles of each cell line were consistent with the corresponding clinical diagnoses and related presentations for the subjects that provided specimens to generate them (Figure 1 and Table 4).

### 3.2. DNA and Protein Analysis of a Cohort of Patients with FMR1 Triplet Repeat Expansions

Having shown the linkage between *FMR1* nucleic acid and protein assays in cell lines immortalized from patients with detailed clinical annotations, we then selected a random group of patients with expanded *FMR1* genotypes that had been assessed through the KKI’s Fragile X Clinic (Clinical Cohort, Table 1). The FM–FMRP Subcohort of the Clinical Cohort was further divided into those with FXS-only and those with FXS + ASD. Additional information on the demographics of this cohort are provided in the Methods section and in Table 1. Whole blood (preserved on FTA cards or 903 cards) and buccal samples (preserved in a stabilizing solution), collected from each subject for subsequent molecular and protein testing, allowed comparisons of *FMR1* DNA and FMRP profiles on cells from different lineages. Since the full set of *FMR1* DNA analyses combined with FMRP quantification were most informative in associating molecular characteristics in FM cases, as shown in our Reference Cohort (Figure 1) and in multiple publications [1,2,4,11,15,19,46], we focused on these measures for subsequent studies.

Using both GS-PCR/CE and RP-PCR/CE, we found that the genotype categories determined for the overall Clinical Cohort were consistent with the original diagnostic test in each case. Specifically, the five PM samples had 56–113 repeats, whereas all 37 FM samples had alleles >200 repeats (Figure 2A,B). Further, the number of CGG repeats was almost always conserved when comparing DNA from whole blood collected in EDTA tubes, whole blood collected on FTA cards, or preserved buccal cells. The differences observed were generally restricted to the appearance of low-level size mosaic peaks (Figure 3 and Figure 4). Overall, slightly more than half of the samples were FM mosaics with evidence of PM alleles. This proportion is roughly in line with previous studies indicating that ~40% of FM samples are allele size mosaics [12,56]. PCR assays that mapped AGG elements in the repeat tract revealed no difference in the interruption pattern when comparing blood and buccal DNA for either unexpanded (most of which had AGG interspersions) or expanded alleles (many of which did not have AGGs).

As expected, the mPCR results indicated partially methylated or unmethylated PM alleles. For example, all three female PM samples were partially methylated and both male PM samples were unmethylated. Further, fully methylated FM alleles were observed in specimens from most subjects with FXS. Indeed, greater than 90% allele-specific methylation was observed for at least one discernible FM peak or near-FM peak in the CE trace for all 37 FM samples. Most samples revealed multiple peaks in the sizeable range of CE; MM was also commonly observed when considering all peaks in a sample, including those with relatively low signal intensity (Figure 3C and Figure 4). Of note, the average methylation of the normal X allele in the six females with FM was 44%, consistent with previous reports [39,47,52] and lyonization, as well as the expectation of a less severe clinical phenotype compared to the 26 FM–FMRP males.

As a next step, we assessed qFMRP levels across all subjects in the Clinical Cohort (Figure 2C). Group analysis of PM compared to FM demonstrated that qFMRP levels were higher in the former; however, three out of five PM were females, who generally have higher qFMRP levels compared to males. In addition, this analysis was limited by having only five individuals with PM. qFMRP levels within the FM–FMRP Subcohort spanned a 30-fold range, when both genders were included (N = 31), and a 12-fold range when only the FM–FMRP male-only subset was analyzed (N = 26). Although our dataset is too limited to definitively untangle the impact of size mosaicism from allele silencing in explaining FMRP expression, we did observe clear trends associating size and MM with FMRP levels. For example, FM alleles only were detected in seven of the ten blood samples with the lowest FMRP levels (i.e., those with less than 2.61 pg FMRP/ng gDNA) in the FM–¬FMRP male Subcohort. Further, these FM alleles were fully methylated in each case. Of the remaining three samples, one had a fully methylated 196 CGG allele and an FM allele, a result consistent with very low FMRP expression (1.2 pg/ng). The other two samples revealed both partially methylated PM and FM alleles; one of them had a 182 CGG PM allele that was only 10% methylated with a fully methylated FM allele. The large size of this PM may have contributed to the low FMRP expression (2.4 pg/ng) in this sample even though the allele was almost completely unmethylated. The remaining sample had a fully methylated FM and 163 CGG allele along with a 32% methylated 122 CGG allele. In this case, the 122 CGG allele may not have been sufficiently abundant in cells and/or efficiently expressed to generate a large amount of FMRP given the low level that was measured (1.48 pg/ng). In contrast to the low-expressing FMRP samples, six of the eight blood samples with FMRP levels higher than the average of 4.17 pg/ng for the Subcohort revealed unmethylated or partially methylated PM alleles.

This analysis shows that epigenetic mosaicism may indicate subpopulations of cells that are competent to express FMRP. Consequently, the FM male-only subset was further analyzed by comparing FMRP levels from subgroups with either appreciable MM (N = 13) or no measurable MM (N = 13) (Appendix A). Of note, the MM subgroup included those with appreciable size mosaicism and MM (as defined in the Methods) but was categorized as MM given the more direct link between methylation status and *FMR1* expression. ANOVA demonstrated that FMRP expression was significantly higher in samples with MM compared to those without MM (*p* = 0.02) (Figure 5).

### 3.3. Blood and Buccal Samples Analyses: Molecular Comparisons within the Overall Clinical Cohort

Heterogeneity in *FMR1* repeat size and methylation occurs in many patients with pathogenic repeat expansions. Nonetheless, it can also manifest in different tissues from the same patient, though the potential phenotypic consequences of these differences are not well understood. Consequently, we compared FM DNA profiles across matched blood and buccal specimens from the same individual. Mosaicism, as we defined it, was (i) detected with high analytical sensitivity using assays previously known to report mosaicism down to 1–5% of cell equivalents [21,57,58] and (ii) detected in about 50% of the subjects in our cohort (Appendix A). As a result, this study offered multiple individuals for comparisons of matched blood and buccal specimens. In general, we found that mosaicism was conserved between these specimens from the same patient. Nevertheless, there were a handful of notable exceptions. One example is a ~101–104 CGG size mosaic in sample 04 that was nearly completely unmethylated in the DNA isolated from EDTA tubes or from the corresponding spotted blood cards but 49% methylated in the buccal DNA (Figure 4A). Another example (sample 15) showed mosaic peaks corresponding to 121 and 162 CGG repeats in blood that were absent in the buccal sample (Figure 4C). When differences in mosaicism were observed, however, they were largely constrained to allele-specific methylation in lower-intensity amplicon peaks by CE; repeat length and even peak intensity of size mosaics was conserved in nearly all samples. Thus, a general finding from our study was the primary genotype and epitype was maintained in matched blood and buccal specimens.

### 3.4. Neurobehavioral Profile of the FM–FMRP Subcohort

Table 5 depicts the neurobehavioral profiles of individuals in the FM–FMRP Subcohort (N = 31). Their age distribution was comparable to the FM–ALL Subcohort (N = 37). Although the number of male and female subjects was not balanced, gender differences in cognitive and adaptive measures, as well as distribution of intellectual functioning, were in line with the literature [35]. The mean FSIQ and adaptive skills composite scores were significantly lower in males than in females (*p* = 0.016 and *p* = 0.001, respectively, Mann–Whitney) and all individuals with moderate or severe ID were males. Frequency of SA diagnosis was high for both genders (68%), especially for females (100%). Unspecified anxiety was only found in males (38%), particularly among the lowest functioning (data not shown). Not surprisingly, ASD was present more frequently in males with FXS (46% vs. 20% females), who also had statistically significantly higher mean ABC-CFX total and CGI-S scores than females (*p* = 0.005 and *p* = 0.036, respectively, Mann–Whitney). FMRP levels were also significantly lower in males than females (*p* = 0.005, Mann–Whitney).

### 3.5. Neurobehavioral Profile of Individuals with ASD in the FM-FMRP Subcohort

In order to determine the influence of ASD status on neurobehavioral profiles in the FM–FMRP Subcohort, we compared males with and without ASD (FXS + ASD and FXS-only). Table 6 depicts these neurobehavioral profiles. Analogous to the differences between males and females, males with ASD had lower level of intellectual functioning, higher proportion of unspecified anxiety diagnosis, and higher mean ABC-CFX total and CGI-S scores than females. FMRP levels were also significantly lower in the FXS + ASD than in the FXS-only group. As expected from the literature, the FXS + ASD group also had higher scores on the ABC-CFX Irritability and Stereotypy subscales and a more prevalent use of antipsychotics. As mentioned above, SA was less prevalent in males FXS + ASD. However, SA severity using a CGI-S adapted measure (CGI-SANX) revealed a closer relationship between SA and ASD. When individuals with mild-moderate CGI-SANX scores (≤4, N = 7) were compared to those with severe CGI-SANX scores (≥5, N = 9), the proportion of males with ASD was higher in the latter (1/7 versus 4/9, respectively). Again, similar to the comparisons between males and females, males with FXS + ASD had lower levels of FMRP than those with FXS-only (Table 6, Figure 6).

### 3.6. Intellectual Functioning, ASD Status, and FMRP Levels

Analyses of the FM–FMRP Subcohort presented above have demonstrated groups with lower cognitive functioning, as well as those with ASD, have lower FMRP levels. Thus, we evaluated in greater detail the relationship between intellectual functioning and ASD, and related behaviors, and FMRP in the male subgroups described in Table 6. First, we examined the distribution of FMRP values in those with or without ASD. There were four outliers in the FXS-only; when either two randomly selected or the four were excluded from the analyses, both parametric and nonparametric *t*-tests showed significant lower levels in the FXS + ASD group (*p* = 0.05 and *p* = 0.01, respectively; Welch’s *t*-test and Mann–Whitney).

Since previous studies have already reported an inverse relationship between intellectual functioning and FMRP levels and a close association between more severe ID and ASD, we also evaluated the role of level of intellectual impairment in the ASD-lower FMRP relationship. An assessment of ID categories and FMRP levels (in pg FMRP/ng gDNA) showed that males with FXS and either mild or moderate ID had significantly higher levels of FMRP than those with severe ID (N = 15, 5.2 ± 3.8 versus N = 10, 2.3 ± 1.1; *p* = 0.013 and *p* = 0.019, respectively; Welch’s *t*-test and Mann–Whitney). These differences were also present when males with FXS + ASD and mild or moderate ID were compared with those with ASD and severe ID (N = 5, 3.3 ± 1.4 versus N = 7, 2.4 ± 1.2) affected by the small subsample (*p* = 0.4, Mann–Whitney), and when similar groups without ASD were compared (FXS-only and mild + moderate ID versus FXS-only and severe ID: N = 10, 6.1 ± 4.3 versus N = 3, 2.1 ± 0.6, *p* = 0. 015 and *p* = 0.07, respectively; Welch’s *t*-test and Mann–Whitney). There was also a statistical trend towards lower FMRP levels in the FXS + ASD subgroup with mild + moderate ID when compared with FXS-only with mild + moderate ID (N = 5, 3.3 ± 1.4 versus N = 10, 6.1 ± 4.3; *p* = 0.075, Welch’s *t*-test), but no differences when groups with severe ID with and without ASD were compared. Another factor that could contribute to the ASD-lower FMRP association is the presence of social behavior impairment that can be interpreted as or associated with ASD. As described above, the group of males with severe SA, by CGI-SANX scores, had a higher proportion of individuals with ASD. This group also had lower FMRP levels than those with mild-moderate SA (3.1 ± 2.2 versus 5.4 ± 3.4; *p* = 0.05, Mann–Whitney).

Finally, we examined the influence of age, overall clinical severity, as measured by CGI-S, and overall problem behavior severity, as determined by ABC-CFX total score, on FMRP levels within the FXS + ASD group. Younger males (2–11 years, 58%) had two-fold lower FMRP levels (pg FMRP/ng gDNA) (N = 7/15, 3.2 ± 1.2 versus N = 8/15, 6.6 ± 4.4 FXS-only, *p* = 0.03, Welch’s *t*-test), adjusted for age- (6.9 ± 0.9 versus 6.9 ± 3.1 FXS-only, *p* = 0.49, Welch’s *t*-test) and FSIQ (44.1 ± 17.7 versus 59.6 ± 16.6 FXS-only, *p* = 0.12, Welch’s *t*-test); while males with an ABC-CFX total score ≥50 and/or severe CGI-S scores (≥5) also closely linked to FXS + ASD, had significantly lower mean levels of FMRP (*p* = 0.01 and *p* = 0.05, respectively; Welch’s *t*-test).To illustrate the potential clinical impact of the *FMR1* and FMRP measurements reported here, one high functioning 7-year old male in the FXS-only group was an outlier with a normal FSIQ of 95 and an adaptive VABS-II Composite score of 79. His FMRP level was 7.4; (mean ± SD) all males (4.2 ± 3.3), FXS-only (5.4 ± 4.0). The subject had unmethylated size and methylation mosaicism on his blood sample and a partially methylated PM smear with an additional PM peak on the buccal sample.

## 4. Discussion

To our knowledge, this is the first study to integrate high-resolution *FMR1* DNA, RNA and protein analyses and to correlate these molecular measures with multiple neurobehavioral parameters, including ASD diagnosis, in order to refine genotype-phenotype correlations in FXS. We first used multiple *FMR1* and FMRP assays to deeply profile pathogenic *FMR1* CGG expansions, and to determine their impact on transcription and translation in well-characterized cell lines from individuals with a range of *FMR1* genotypes. We then applied the most revealing of these assays to an FXS cohort with comprehensive neurobehavioral profiling to investigate molecular profiles and molecular-neurobehavioral correlations.

Using a combination of sensitive and quantitative multi-omic assays, we found that *FMR1* CGG repeat length, methylation levels, and FMRP levels were complementary and consistent with our current knowledge of *FMR1* translational biology in individuals with normal, PM, and FM alleles [1,2,4,5]. FM expansions were associated with hypermethylation, gene silencing based on *FMR1* mRNA levels much lower in individuals with FM than carriers with PM, and FMRP levels lower in males with FM than in females with FM and much lower in subjects with FM overall compared to individuals with PM. Molecular profiles in lymphoblastoid cell lines were corroborated in blood samples with a variety of *FMR1* expansions; these expansions included a subject with an FM allele that was unmethylated in both immortalized cell line and original blood specimen and exhibited *FMR1* mRNA and FMRP levels similar to those of individuals with PM. Although unmethylated FM alleles are rare, several examples have been described in the primary literature [2,4,15,16,39]. We also observed that blood-based profiles were in general consistent with those in buccal epithelial specimens. These findings are contributory since concordance in patterns of *FMR1* mosaicism across tissues continues to be a controversial issue in FXS [46,59,60]. Moreover, very little is known about the correlation between blood FMRP measurements and postmortem brain tissue FMRP levels in the same subject [61,62]. Only a single case study “verified” reduced expression of *FMR1* mRNA and FMRP in both peripheral blood and brain leading to the FXS [61]. In our study, a representative sample of males and females with FXS, with their corresponding range of cognitive and behavioral impairments [9,29], served as the basis for examining *FMR1* methylation, FMRP levels and their relationship with neurobehavioral features. As expected, samples with complete or near complete *FMR1* gene methylation by mPCR exhibited the lowest FMRP values. Females with FXS, not surprisingly, showed lower relative gene methylation and higher FMRP levels. FMRP levels were generally lower for males with FM-only compared to males with PM, females with FM, and males with methylation mosaicism, with the latter groups displaying a wide range of values as described earlier [9,10,19]. A significant finding from this study is that FMRP levels were two-fold higher in males with FXS and appreciable methylation mosaicism compared to males without such mosaicism, expanding our knowledge on the link between methylation and *FMR1* silencing. In other published work, Jiraanont and colleagues used *FMR1* PCR and mPCR along with Southern blot, RT-qPCR, and FMRP levels to evaluate size- and methylation-mosaicism in 12 males with FXS (7 with FXS + ASD) [11]. They concluded lower *FMR1* mRNA and FMRP levels were the main contributors to cognitive impairment and the presence of a normal allele appeared to compensate in some but not all individuals. Pretto et al. [46] studied 18 patients with FXS (9 with FXS + ASD), including 7 with size- and 6 with methylation-mosaicism, demonstrating that *FMR1* RNA and FMRP levels correlated in general with *FMR1* methylation in peripheral blood cells and fibroblasts although there were differences in CGG expansion and blood samples showed lower methylation. They also reported FSIQ scores were inversely correlated with level of methylation and directly with FMRP levels. However, other aspects of the neurobehavioral phenotype, such as number of seizures and severity of hyperactivity or autistic behavior were not related to methylation or FMRP levels.

Considering the relatively consistent relationship between *FMR1* methylation, *FMR1* mRNA, and FMRP and the more direct phenotypical implications of protein expression, FMRP levels seem to be the most suitable molecular parameter for molecular-neurobehavioral analyses [1,2]. Our findings confirm previous studies [9,10,29,36] showing FMRP is a biomarker of overall clinical severity in FXS, which may help stratify patients with FXS in clinical trials [38]. We found a strong link between FMRP levels and IQ or level of ID [1,7,9,10,11,19,29,36,46], evident in male–female comparisons and in analysis of males with a range of clinical presentations. Nonetheless, the availability of a sensitive and precise assay such as qFMRP [43], allowed us to detect other phenotypical correlates of FMRP deficiency not reported in the aforementioned studies. For instance, we found two-fold lower levels of FMRP particularly in younger males (age- and IQ-adjusted) with FXS and ASD (57% with severe ASD) than in those with FXS-only, although there was a wide range of protein expression in the latter group. Nonetheless, for total FM–FMRP male-only subset, this finding only held in ones with mild-moderate ID, as those with severe ID had even lower FMRP levels independent of ASD status. Since marked decrease in FMRP is not only the basis for FXS but also a contributor to the ASD phenotype, as the signaling pathways of *FMR1* and other genes linked to the behavioral disorder substantially overlap [23,24,28,63], a more definitive answer about the relationship between FMRP levels and ASD status deserves further study. Other relationships between FMRP levels and neurobehavioral parameters included unspecified anxiety only in low functioning males with low FMRP and ASD, and SA in females and higher functioning males without ASD but with higher FMRP levels. Overall, our study replicated an extremely high rate of anxiety in FXS reported in the literature regardless of gender [64]. Anxiety is intertwined in FXS [64,65], and in FXS with ASD [36]. Two potentially clinically meaningful SA subgroups within the male-only group were identified with separately applied CGI-S SA scoring (CGI-SANX)—an approach which has also been used in a second wave of clinical trials in FXS (i.e., the NCT03697161gaboxadol recently completed study). Further study of anxiety in relationship to FMRP and ASD status in FXS may also help stratify patients with FXS in clinical trials [38].

As with ASD in males with severe ID, the relationship between FMRP levels and overall severe problem behavior (ABC-CFX total score ≥50) was also cofounded with greater cognitive impairment. These findings are in line with the recognition, in recent years, of a more severe neurobehavioral phenotype mainly in males with FXS. This is characterized by severe ID, ASD diagnosis, and severe irritability, aggression, agitation, and self-injury types of often anxiety-driven behavior [25,27,66]. Indeed, our sample of males with FXS and ASD [26,34] had significantly elevated ABC-CFX total scores driven by the Irritability subscale [67], and a higher proportion of anxiety diagnosis and CGI-SANX scores, respectively, which together required frequent use of atypical antipsychotic aripiprazole [68,69,70]. A presumed link between anxiety and ABC-CFX Irritability subscale in lower functioning individuals with FXS also led experts in fragile X field to include the latter as an outcome measure in cannabinol clinical trials in FXS (NCT03614663 and NCT03802799). Our findings suggest that the lowest FMRP levels seem to concentrate in this group of male individuals. Availability of sensitive assays like qFMRP used here are key for advancing our understanding of the role of FMRP deficit in FXS, since threshold levels (<70% of FMRP level observed in those with normal CGG repeat numbers) for ID have been recently described [19] and delineating differences among those with the lowest levels will require high power discrimination [38,53,71].

## 5. Limitations

Although our study, to our knowledge, is the largest molecular–neurobehavioral correlation using multiple and sensitive *FMR1* and FMRP assays, the sample size was relatively modest considering the variability and complexity of the molecular and phenotypical measures. Marked neurobehavioral differences between males and females with FXS, coupled with the well-known genetic differences inherent to X-linked disorders (e.g., X-inactivation in females) complicated the molecular and correlational analyses. Given the size of the cohort, the necessity of combining size- and methylation-mosaicism limited the full appreciation of differences between these types of mosaicism. Despite the use of raw score-based z-score calculations for FSIQ, floor effects on cognitive or adaptive skills testing for some subjects were also a limitation for studying range of cognitive impairment. The lack of a formal anxiety measure also limited our ability to subtype anxiety.

## 6. Conclusions

Our findings underscore the association between *FMR1* CGG expansion, gene methylation, FMRP levels, and overall neurobehavioral severity, including an ASD diagnosis. The synergy of *FMR1* genomic and protein parameters and the concordance of blood and buccal profiles using a mixture of sensitive and quantitative multi-omic assays support the benefit of characterizing molecular profiles particularly in males with FXS in observational and interventional studies. The benefit of stratifying their genotype–phenotype profiles is further strengthened by two-fold higher FMRP levels and lesser clinical severity in a subset of these males with substantial methylation mosaicism. Thus, the benefit also highlights the potential of FMRP levels, in particular, as a prognostic marker in clinical trials and other relevant studies in males with FXS.

## Figures and Tables

**Figure 1 brainsci-10-00694-f001:**
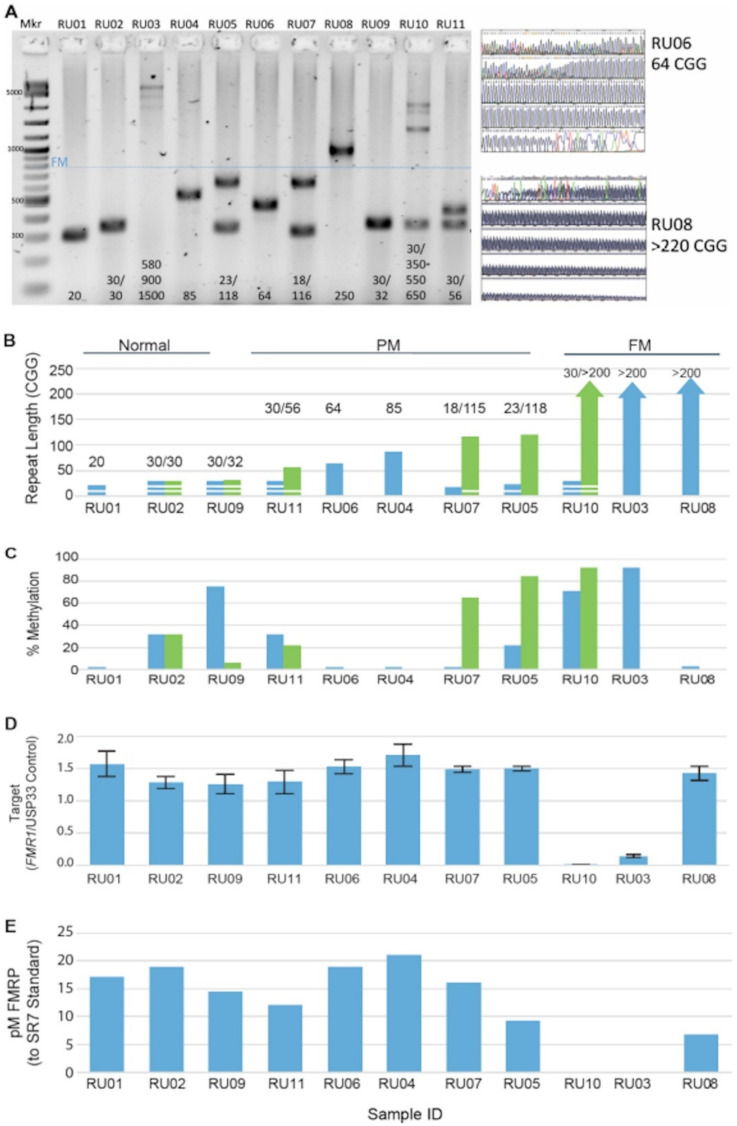
(**A**) GS-PCR/AGE and Sanger sequencing for repeat sizing. (**B**) RP-PCR/CE repeat sizing and AGG interruptions (represented as horizontal column breaks, determined using Xpansion Interpreter PCR). (**C**) Methylation percentage assessed with mPCR. (**D**) *FMR1* mRNA expression levels for matched cell lines (RT-qPCR), and (**E**) FMRP levels normalized to the recombinant FMRP peptide GST-SR7. In (**B**,**C**), the second (usually longer) CGG allele for female cell lines is shown in green; all others are given in blue.

**Figure 2 brainsci-10-00694-f002:**
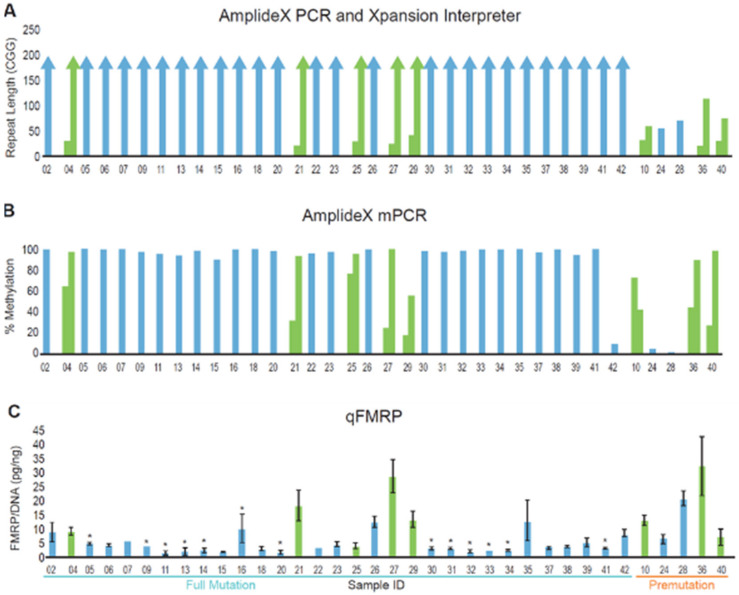
(**A**) *FMR1* genotype (top panel) and (**B**) methylation status (middle panel) correlated with (**C**) FMRP levels (lower panel). Samples were annotated for *FMR1* CGG repeat length and degree of methylation using AmplideX PCR/CE *FMR1* and mPCR technologies. Male and female samples are colored in blue and green, respectively. Bottom panel: Error bars represent standard deviations. FM samples without methylation mosaicism are indicated with an asterisk above the bar.

**Figure 3 brainsci-10-00694-f003:**
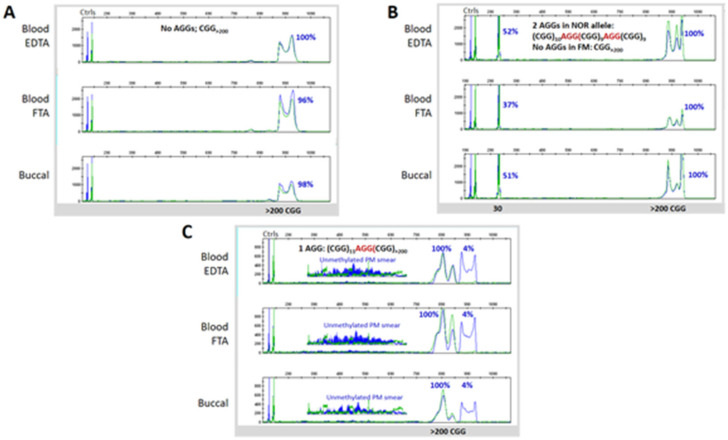
Blood samples collected in EDTA tubes or spotted on FTA paper used only for DNA analyses were compared with cheek swab specimens. DNA analyses from three representative FM subjects are shown. (**A**) Male with FXS (sample 09) with a fully methylated expansion and no detectable size or methylation mosaicism. (**B**) Female with FXS (sample 19) with random X-inactivation of the normal allele and a fully methylated FM allele. (**C**) Male with FXS (sample 02) with FM, PM size and methylation mosaicism in both blood and buccal gDNA. Capillary electrophoresis traces show CGG repeat length (blue, undigested) and methylation status (green, digested).

**Figure 4 brainsci-10-00694-f004:**
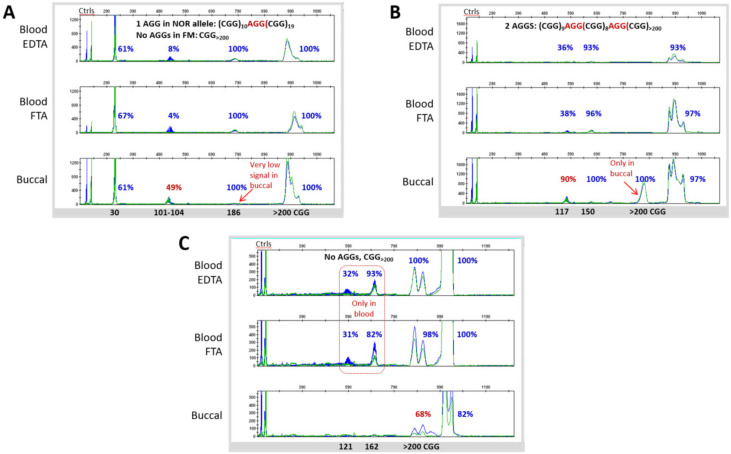
Level of concordance in primary allele CGG sizing and methylation status in blood and buccal specimen from subjects with FXS. (**A**) Female with FXS (sample 04) with PM size mosaicism and differential relative methylation in blood and buccal gDNA. (**B**) Male with FXS (sample 06) with an additional FM peak in buccal cells compared to blood, along with methylation mosaicism. (**C**) Male with FXS (sample 15) with a blood-only PM size mosaicism and reduced relative methylation of the FM in the buccal sample. Capillary electrophoresis traces show CGG repeat length (blue, undigested) and methylation status (green, digested).

**Figure 5 brainsci-10-00694-f005:**
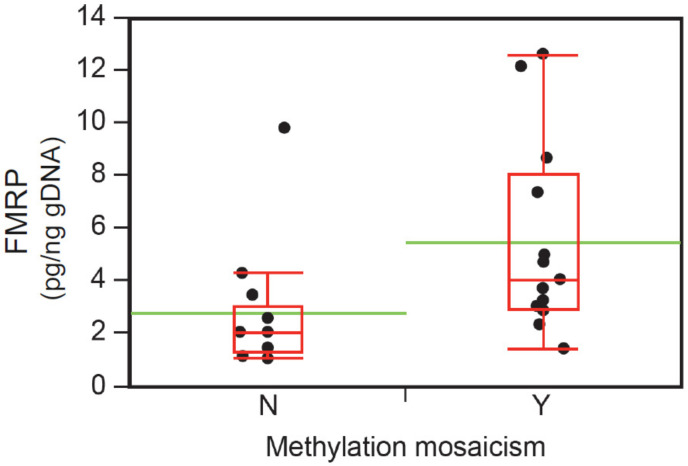
FMRP levels and methylation mosaicism (MM) status from whole blood specimens. In the FM–FMRP Subcohort males (N = 26), those with MM (Y) had significantly higher FMRP levels as measured by the qFMRP assay. One-way ANOVA (*p* = 0.02), where the green line shows the mean for each group, and box plots represent the median and quantiles. Y (yes), N (no) methylation mosaicism.

**Figure 6 brainsci-10-00694-f006:**
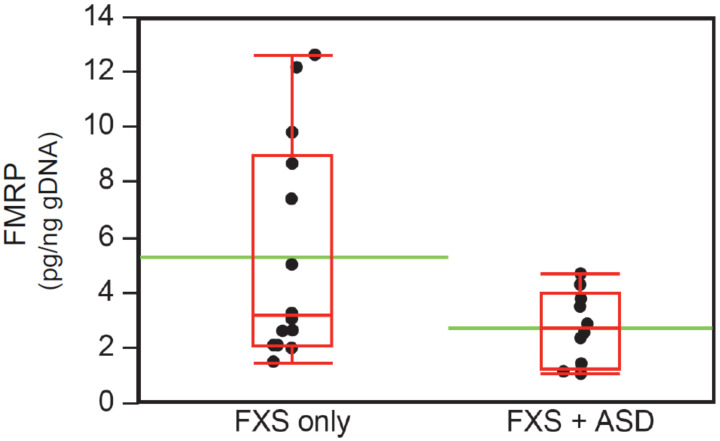
FMRP levels in whole blood specimens from males with FXS with and without ASD. qFMRP measurements showed significantly higher levels in the FXS-only group than in those with FXS + ASD (*p* = 0.04, Welch’s *t*-test). The box plots outline median and percentiles 25 and 75 for each group, while the green line represents the mean for each group.

**Table 1 brainsci-10-00694-t001:** Characteristics of the clinical cohort.

	Clinical Cohort		FM–ALL Subcohort		FM–FMRP Subcohort *	
	N = 42		N = 37		N = 31	
Gender	Male	Female	Male	Female	Male	Female
N	33	9	31	6	26	5
Age years						
Mean (SD)	14.4 (11.9)	14.7 (10.9)	14.7 (12.6)	11.8(7.3)	13.8 (11.9)	13.2 (7.2)
Median	10	10	10	10	9.5	10
Range	2.6–47.1	5–39	2.6-47.1	5–26	2.6-47.1	9–26
Race, N						
Caucasian	25	8	23	5	20	4
African American	3	1	3	1	2	1
Asian	4	/	4	/	3	/
Hispanic	1	/	1	/	1	/
*FMR1* expansions, N						
FM	31	6	31	6	26	5
PM	2	3	/	/	/	/

Abbreviations: FM—full mutation, FMRP—fragile X mental retardation protein, N—number of participants, PM—premutation, SD—standard deviation. * Detailed clinical neurobehavioral assessments of the FM–FMRP Subcohort are presented in Table 3.

**Table 2 brainsci-10-00694-t002:** Summary of samples and molecular analyses.

Materials	DNA Genotype Repeat Size Analysis	DNA Epitype Methylation Analysis	RNA Expression Analysis	Protein Quantification
**Reference Cohort** Cell lines (11/11)	GS-PCR/CE GS-PCR/AGE RP-PCR/CE AGG interruption Sanger Sequencing Southern Blot	mPCR	RT-qPCR	qFMRP
**Clinical Cohort** WB (20/42) FTA cards (42/42) 903 protein saver cards (36/42) Buccal (42/42)	GS-PCR/CE RP-PCR/CE AGG interruption	mPCR		qFMRP (36/42)

The number of subjects or cell lines tested is given in parentheses. Abbreviations: WB—whole liquid blood, FTA—FTA blood spot cards, GS-PCR/CE—Gene-Specific PCR/CE, GS-PCR/AGE—Gene-Specific PCR/Agarose Gel Electrophoresis, RP-PCR/CE—Repeat-Primed PCR/CE, AGG—adenine-guanine-guanine, mPCR—methylation PCR, RT-qPCR—Reverse transcription-quantitative PCR, qFMRP—quantitative FMRP.

**Table 3 brainsci-10-00694-t003:** Neurobehavioral assessments of the FM–FMRP subcohort.

Domain	Measure	Categories
Diagnosis	DSM-5 criteria	ASD *
		Anxiety **
Intellectual Functioning	FSIQ	Normal range
		Mild ID
		Moderate ID
		Severe ID
Problem Behaviors (Parent report)	ABC-C_FX_	(i) Irritability
		(ii) Lethargy/Social Withdrawal
		(iii) Stereotypic Behavior
		(iv) Hyperactivity
		(v) Inappropriate Speech
		(vi) Social Avoidance
Overall Clinical Severity	CGI-S	Severity Score Range
		Not at all ill (1)
		To
		Extremely ill (7)
Antipsychotic Use	Health Records	Yes
		No

Abbreviations: DSM-5—Diagnostic and Statistical Manual, 5th edition; ASD—autism spectrum disorder; FSIQ—full scale intellectual quotient; ABC-C_FX_—Aberrant Behavior Checklist-Community (fragile X version); CGI-S—clinical global impression-severity. * Subcohort also received Autism Diagnostic Observation Schedule (ADOS) assessments. ** Includes social and unspecified anxiety diagnoses. A separate CGI-S scoring approach was also applied to profile for severity of social anxiety diagnosis (CGI-SANX).

**Table 4 brainsci-10-00694-t004:** Genetic and clinical characteristics of 11 subjects who provided blood cells for immortalized lymphoblastoid cell lines used in this study.

Subject	Gender	CGG Repeat	Clinical Annotation	CGG Repeat
		Genotype *		Genotype ** Cell Line
		Whole Blood		
RU01	M	20	Clinically normal	20
RU02	F	30/30	Clinically normal	30/30
RU03	M	>200 (~900)	FXS and moderate intellectual impairment, fully methylated FM	>200
RU04	M	78	Clinically unaffected, Grandchild with FXS, unmethylated PM allele	85
RU05	F	23/118, 124–169, minor >200	Chronic fatigue syndrome, Daughter with FXS, with low-level mosaicism for an FM allele of >200 repeats. Both methylated and unmethylated expanded alleles apparent by Southern blot analysis.	23/113, 118, FM not present
RU06	M	62	Clinically unaffected Grandson with FXS unmethylated PM	64 41, 51
RU07	F	18/116 (>200)	Mild tremor, Son with FXS PM allele with repeat length of 115 with low-level mosaicism and an FM allele of >200 repeats. Both methylated and unmethylated expanded alleles apparent by Southern blot analysis.	18/115
RU08	M	~180 mosaicism and >200	Male with normal development but mild symptoms of FXTAS and grandchildren with FXS, donor subject has alleles with unmethylated CGG repeat lengths of about 180 and >200 by Southern blot analysis.	>200, 86
RU09	F	30/32	Clinically normal	30/32
RU10	F	30/>200	Female with FXS and mild ID, a fully methylated FM allele with CGG repeat length of ~600	30/>200
RU11	F	30/56	Clinically unaffected relatives with FXS	30/56

* Genotype determined using *FMR1*-specific repeat-primed PCR (RP-PCR) or a combination of RP-PCR and Southern blot analysis. Values are given as the number of CGG repeats and include minor alleles; ** Genotype determined from cell lines with RP-PCR assay FXTAS—fragile X-associated tremor ataxia syndrome.

**Table 5 brainsci-10-00694-t005:** Neurobehavioral profiles by gender in the FM–FMRP Subcohort.

Variable	N	FM–FMRP Subcohort		^§^ F (df)/	*p*
		N = 31		^†^ Chi-Square (df)	
		Males	Females		
		N = 26	N = 5		
Age years, mean (SD)	31	14.0 (11.9)	13.2 (7.2)	0.01 (1) ^§^	0.58
FSIQ, mean (SD)	31	43.6 (18.3)	74.8 (19.1)	1.36 (1) ^§^	0.016 *
Adaptive Skills Composite, mean (SD)	31	60.8 (12.0)	79.6 (13.2)	11.47 (1) ^§^	0.001 *
Intellectual functioning ^¥^, N	31				
Normal range IQ, N (% of N)	3	1 (4)	2 (40)	35.69 (1) ^†^	<0.0001 ^†^*
Mild ID, N (% of N)	7	4 (16)	3 (60)	39.24 (1) ^†^	<0.0001 ^†^*
Moderate ID, N (% of N)	11	11 (42)	0 (0)	50.66 (1) ^†^	<0.0001 ^†^*
Severe ID (*%* of N)	10	10 (38)	0 (0)	44.48 (1) ^†^	<0.0001 ^†^*
CGI-S (overall), mean (SD)	31	5.0 (0.9)	4.0 (0.7)	5.7 (1) ^§^	0.036*
Anxiety, total N	31				
Social Anxiety, N (% of N)	21	16 (62)	5 (100)	0.24 (1) ^†^	0.63
Unspecified anxiety, N (% of N)	10	10 (38)	0 (0)	0.24 (1) ^†^	0.63
ASD, N (%)	31	12 (46)	1 (20)	1.18 (1) ^†^	0.28
ABC-C_FX_, mean (SD)	30	59 (19.6)	28.7 (12.6)	14.0 (1) ^§^	0.005 *
Antipsychotics, N (% of N)	31	10 (38)	0 (0)	0.8 (1) ^†^	0.09
FMRP, mean pg/ng (SD)	31	4.2 (3.3)	14.3 (9.5)	19.7 (1) ^§^	0.005 *

Abbreviations: FM—full mutation, ASD—autism spectrum disorder, FSIQ—full scale intelligence quotient, raw-score based z-score calculations from the IQ subtests as the basis for an extended FSIQ values, IQ—intelligence quotient, ID—intellectual disability, CGI-S—Clinical Global Impression-Severity, ABC-CFX—Aberrant Behavior Checklist-Community (fragile X version), FMRP—fragile X mental retardation protein, N—number of participants, SD—standard deviation, ^§^ F—statistic ratio for one-way ANOVA, df—degrees of freedom. ^†^ Chi-Square test was used, * statistically significant continuous data, Mann–Whitney, ^†^* Chi-Square statistically significant ^¥^ Chi-Square Intellectual functioning: normal range IQ (FSIQ score > 70); mild ID (FSIQ: 55–69), and moderate (FSIQ: 35–54) and severe ID (FSIQ < 35). FMRP levels are provided in pg FMRP per ng of gDNA to normalize for cell count.

**Table 6 brainsci-10-00694-t006:** Neurobehavioral profiles by ASD status in males in the FM–FMRP Subcohort.

	N	FM–FMRP Males		^§^ F (df)/	*p*
		*N* = 26		^†^ Chi-Square (df)	
		FXS-Only	FXS + ASD		
		N = 14	N = 12		
Age in years, mean (SD)	26	17.1 (15.1)	9.9 (4.9)	2.5 (1) ^§^	0.13
FSIQ, mean (SD)	26	49.8 (17.5)	36.4 (17.0)	3.6 (1) ^§^	0.03 *
Adaptive Skills Composite, mean (SD)	26	62.3 (13.9)	58.2 (10.5)	1.8 (1) ^§^	0.17
Intellectual functioning ^¥^, N	26				
Normal range IQ, N (% of N)	1	1 (7)	0 (0)	5.3 (1) ^†^	0.02 ^†^*
Mild ID, N (% of N)	4	3 (21)	1 (8)	5.8 (1) ^†^	0.01 ^†^*
Moderate ID, N (*%* of N)	11	7 (50)	4 (33)	5.3 (1) ^†^	0.02 ^†^*
Severe ID, N (% of N)	10	3 (21)	7 (58)	27.1 (1) ^†^	<0.0001 ^†^*
CGI-S (overall), mean (SD)	26	4.6 (0.9)	5.9 (0.7)	6.6 (1) ^§^	0.03 ^†^*
Anxiety, N	26				
Social Anxiety, N (%)	16	11 (79)	5 (42)	7.9 (1) ^†^	0.05 ^†^*
Unspecified Anxiety, N (%)	10	3 (21)	7 (58)	7.9 (1) ^†^	0.05 ^†^*
ABC-C_FX_, mean (SD)	26	51.7 (17.0)	67.4 (19.7)	4.8 (1) ^§^	0.04 *
ABC Irritability	26	17.2 (9.2)	26 (11.7)	4.6 (1) ^§^	0.04 *
ABC Unresponsive Lethargy	26	6.8 (4.6)	8.7 (3.7)	1.4 (1) ^§^	0.2
ABC Stereotypy	26	6.6 (3.8)	10.2 (4.7)	4.6 (1) ^§^	0.04 *
ABC Hyperactivity	26	15.4 (8.6)	18.8 (8.1)	1.1 (1) ^§^	0.3
ABC Inappropriate Speech	26	4.8 (3.3)	5.8 (3.7)	0.6 (1) ^§^	0.4
ABC Social Avoid	26	3.9 (3.0)	4.0 (2.6)	0.02 (1) ^§^	0.9
Antipsychotics, N (%)	26	3 (21)	7 (58)	3.7 (1) ^†^	0.05 ^†^*
FMRP [pg/ng], mean (SD)	26	5.4 (4.0)	2.8 (1.3)	4.6 (1) ^§^	0.04 *

Abbreviations: FM—full mutation, FXS—fragile X syndrome, ASD—autism spectrum disorder, FSIQ—full scale intelligence quotient, Extended raw-score based scoring of the FSIQ, ID—intellectual disability, CGI-S—Clinical Global Impression-Severity, ABC-CFX—Aberrant Behavior Checklist-Community (fragile X version), FMRP—fragile X mental retardation protein, N—number of participants, SD—standard deviation, ^§^ F—statistic ratio for one-way ANOVA, df—degrees of freedom. ^†^ Chi-Square test, * statistically significant continuous data with randomly selected FXS-only outliers included, Welch’s *t*-test, ^†^* Chi-Square statistically significant; ^¥^ Intellectual functioning: normal (FSIQ score > 70); mild ID (FSIQ: 55–69), moderate (FSIQ: 35–54) and severe ID (FSIQ < 35). FMRP levels are provided in pg FMRP per ng of gDNA to normalize for cell count.

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
