# Peer review of "A Genotype-Phenotype Study of High-Resolution FMR1 Nucleic Acid and Protein Analyses in Fragile X Patients with Neurobehavioral Assessments"

_brainsci, 2020, doi:10.3390/brainsci10100694_

Round 1

Reviewer 1 Report

This manuscript describes a detailed analysis of DNA, mRNA, and FMRP protein in 31-42 subjects with full mutation FXS, and neurobehavioral assessments on 31 of the subjects.  Although previous studies have reported some data on these parameters, the strength of this study derives from the comprehensiveness of the molecular and clinical measurements on the same subjects, and the use of new improved analytical techniques for the molecular measurements.  The paper is well written and the findings reveal a useful consolidation of the relationships between FMR1 genotype, methylation status, mRNA expression, and FMRP levels, and their relationship with general cognitive and emotional status of the subjects.  However, the manuscript requires clarification and further information on several points as outlined below.

Abstract

  1. FMR1CGG repeat expansions, methylationlevels, and FMRP levels, in both cell lines and blood samples, were consistent with previous FMR1genomic and protein studies.” 

How or consistent with what?

Methods

  1. More details should be added to the methods relating to the subsection on Quantitative FMRP analysis, including how the protein was extracted from the samples, how much protein was used in the assay, and a clarification of the phrases “(on 903 paper)” and “FTA paper”.

Results and Figures

  1. Figure 1 legend should state what is represented by the green and blue colors. In Figs. 5 and 6 the legends should state that FMRP analysis is from whole blood samples.
  2. In the files that I examined, some of the graphics were too small and not clearly visible, even after enlargement. This may partially be an issue of the file type, .pdf, but it in several figures e.g. Figs. 3 and 4 the annotation within the figure may be too small, even with higher resolution.
  3. “However, one FM cell-line (RU08) was unmethylated (Fig. 1C).” It would be useful to add a statement in the Results or Discussion, based on previous studies, as to how common this situation is in the FM population (e.g. roughly what per cent of the general FM population falls into this category?).
  4. In the results section in the paragraph beginning “Finally, we measured FMRP levels for each cell-line…”, it would be helpful to state here (again?), what the baseline was.
  5. In table 6, it should be stated how the 2-4 outliers were defined and whether or not they were included in the results shown in table 6 (probably yes).
  6. Several of the tables include information on the use of antipsychotics by the subjects, but then no further statements are made relating to this. Considering that a wide array of drugs is used to “treat” FXS, why is there a mention of only this class of drugs?  Did the authors examine this further, and can any conclusion or statement be made as to the significance or relevance of antipsychotic use in the context of this study?
  7. Why did the authors not examine or report how FMRP levels correlated with IQ or ID or some other behavioral parameter across all subjects and/or across subsets of subjects (i.e. correlation coefficients)?

Discussion

  1. “A significant finding from this study is that FMRP levels were two-fold higher in males with FXS and appreciable methylation mosaicism, expanding our knowledge on the link between methylation and FMR1 silencing.’ Two-fold higher compared to what, without MM?
  2. “Availability of sensitive assays like qFMRP used here arekey for advancing our understanding of the role of FMRP deficit in FXS, since threshold levels (<70%  typical) for ID have been recently described [19] and delineating differences among those with the lowest levels will require high power discrimination.”  What does the term “threshold levels (<70% typical)” refer to – IQ, or FMRP?  If the latter, how is the threshold level determined or established?
  3. The authors emphasize the utility of measuring FMRP. At the risk of opening a proverbial can of worms, could the authors add a few sentences summarizing (with references) what is known about the correlation between blood FMRP measurements and post mortem brain tissue FMRP levels in the same subject?

Minor comments

Typos “onsistent with 

 “….previous reports [39, 47, 52] and lyonization,”

In Table 5 it looks like a label for the first column is missing (N ?).

Author Response

This manuscript describes a detailed analysis of DNA, mRNA, and FMRP protein in 31-42 subjects with full mutation FXS, and neurobehavioral assessments on 31 of the subjects.  Although previous studies have reported some data on these parameters, the strength of this study derives from the comprehensiveness of the molecular and clinical measurements on the same subjects, and the use of new improved analytical techniques for the molecular measurements.  The paper is well written and the findings reveal a useful consolidation of the relationships between FMR1 genotype, methylation status, mRNA expression, and FMRP levels, and their relationship with general cognitive and emotional status of the subjects.  However, the manuscript requires clarification and further information on several points as outlined below.

We are grateful to reviewer 1 for thoughtful comments that we addressed below point by point.

Abstract

  1. FMR1 CGG repeat expansions, methylation levels, and FMRP levels, in both cell lines and blood samples, were consistent with previous FMR1 genomic and protein studies.” 

How or consistent with what?

Response: We are thankful for the reviewer’s comment.  We clarified this sentence in Abstract (Line 48) to read as follows.

“FMR1 CGG repeat expansions, methylation levels, and FMRP levels, in both cell lines and blood samples, were consistent with findings of previous FMR1 genomic and protein studies.” 

In addition, this sentence also appears in the Discussion section (Lines 592-595) with several relevant references (1,2,4,5).

“Using a combination of sensitive and quantitative multi-omic assays we found that FMR1 CGG repeat length, methylation levels, and FMRP levels were complementary and consistent with our current knowledge of FMR1 translational biology in individuals with normal, PM, and FM alleles [1, 2, 4, 5].”

These references, and the studies they also cite, provide a foundation to compare our characterizations of DNA, RNA and protein, both relative to one another and with the primary clinical presentation by each subject. For example, FM expansions were associated with methylation silencing, FMR1 mRNA levels were much lower in silenced FM individuals than PM carriers, and FMRP levels were lower in FM males than FM females and much lower in FM subjects overall compared to PM individuals.  These references also acknowledge rare molecular profiles that we describe in this study (such as unmethylated FM alleles).   

Thus, we also expanded on this context to the Discussion in Lines 595-598. FM expansions were associated with hypermethylation, gene silencing based on FMR1 mRNA levels much lower in individuals with FM than carriers with PM, and FMRP levels lower in males with FM than in females with FM and much lower in subjects with FM overall compared to individuals with PM.

Methods

  1. More details should be added to the methods relating to the subsection on Quantitative FMRP analysis, including how the protein was extracted from the samples, how much protein was used in the assay, and a clarification of the phrases “(on 903 paper)” and “FTA paper”.

Response: We are thankful for the reviewer’s comment. To clarify these methods, we combined and expanded the descriptions of protein isolation and qFMRP quantification into the same Methods section; see lines 258-271.

Lines 258-271: “Briefly, for the protein extraction, eight 3 mm punches from blood spotted and dried on Whatman 903 paper were added to 200 ul M-PER solution (Thermo Fisher Scientific, Waltham, MA) supplemented with protease inhibitors (cOmplete, Mini, EDTA-free; Roche Applied Science, Indianapolis, IN). After incubation for 3 hrs at room temperature, samples were briefly centrifuged at 10,000 x g to remove cell debris.  Two 50 ul aliquots of the eluate were independently analyzed in the qFMRP assay using liquid bead array (Luminex, Austin, TX) following the method described by Gustin et al. [42].  The FMRP concentration in each sample was determined after comparing the relative fluorescent intensity against an 11-point standard curve constructed using 0.28 to 280 pM of the GST-SR7 peptide, along with a blank.  To normalize FMRP concentration across blood specimens, gDNA was used as a proxy for cell count.  For each sample, gDNA was quantified using the PicoGreen Quant-iT™ HS kit (Thermo Fisher Scientific, Waltham, MA) and the FMRP level standardized to the gDNA amount.  Where relevant, the total protein concentration of the lysates was determined using a Pierce BCA Protein Assay Kit (Thermo Fisher Scientific, Waltham, MA).”

Results and Figures

  1. 3. Figure 1 legend should state what is represented by the green and blue colors. In Figs. 5 and 6 the legends should state that FMRP analysis is from whole blood samples.

Response: We are thankful for the reviewer’s comment. The figure legends for Figs. 1, 5, and 6 have been revised accordingly.

  1. 4. In the files that I examined, some of the graphics were too small and not clearly visible, even after enlargement. This may partially be an issue of the file type, .pdf, but it in several figures e.g. Figs. 3 and 4 the annotation within the figure may be too small, even with higher resolution.

Response: We are grateful for the reviewer’s comment. We have enlarged and provided higher resolution, more legible versions of Figs. 3 and 4 with the revised manuscript.

  1. “However, one FM cell-line (RU08) was unmethylated (Fig. 1C).” It would be useful to add a statement in the Results or Discussion, based on previous studies, as to how common this situation is in the FM population (e.g. roughly what per cent of the general FM population falls into this category?).

Response: We are thankful for the reviewer’s comment. We have elaborated of this rare unmethylated FM sample in our Discussion section, and provided literature references to help contextualize similar findings from the literature.  See lines 410-412 already in the body text, and revised text in lines 595-598 and lines 598-602. 

Lines 410-412 Overall, slightly more than half of the samples were FM mosaics with evidence of PM alleles. This proportion is roughly in line with previous studies indicating that ~40% of FM samples are allele size mosaics [12, 56].

Lines 595-598: “FM expansions were associated with hypermethylation, gene silencing based on FMR1 mRNA levels much lower in individuals with FM than carriers with PM, and FMRP levels lower in males with FM than in females with FM and much lower in subjects with FM overall compared to individuals with PM.”

Lines: 598-602: “Molecular profiles in lymphoblastoid cell lines were corroborated in blood samples with a variety of FMR1 expansions; these expansions included a subject with a FM allele that was unmethylated in both immortalized cell line and original blood specimen and exhibited FMR1 mRNA and FMRP levels similar to those of individuals with PM.  Although unmethylated FM alleles are rare, several examples have been described in the primary literature [2, 4, 15, 16, 39].

  1. In the results section in the paragraph beginning “Finally, we measured FMRP levels for each cell-line…”, it would be helpful to state here (again?), what the baseline was.

Response: We are thankful for the reviewer’s comment. We revised this passage in line 372 and lines 376-378 to state that the comparison across samples was assessed against the limit of detection for the assay.  The FMRP levels in two samples, RU03 and RU10, failed to rise above this threshold.

Lines 372 and 376-378: “Finally, we measured FMRP levels for each cell line to assess the impact of repeat expansions and other FMR1 DNA and RNA parameters on translation. Lysates of the 11 cell lines were analyzed by a previously established antibody-based qFMRP liquid bead array assay [42, 43]. As shown in Fig. 1E, FMRP levels were found to be well above the the limit of detection for nine of the 11 cell lines including the unmethylated FM cell line (RU08). The two FM samples (RU09 and RU10) with pronounced gene methylation showed absence of FMRP. Thus, the results indicate that this set of multianalyte assays can quantify FMR1 gene and FMRP parameters across the range of FMR1 categorical genotypes with complementary results. Further, the molecular profiles of each cell line were consistent with the corresponding clinical diagnoses and related presentations for the subjects that provided specimens to generate them (Fig. 1 and Table 4).”

  1. In table 6, it should be stated how the 2-4 outliers were defined and whether or not they were included in the results shown in table 6 (probably yes).

Response: We are thankful for the reviewer’s comment.

In the FXS-only (N = 14) group the 2-4 outliers were all included in Table 6. The outliers were randomly selected by SPSS-program. We added on such clarification in the Table 6.

Line 533-534 *statistically significant continuous data with randomly selected FXS-only outliers included, Welch’s t-test.

In addition, right below Table 6 in the body text (lines 541-544), we already stated that “There were four outliers in the FXS-only; when either two randomly selected or the four were excluded from the analyses, both parametric and nonparametric t-tests showed significant lower levels in the FXS + ASD group (p = 0.05 and p = 0.01, respectively; Welch’s t-test and Mann-Whitney).

  1. Several of the tables include information on the use of antipsychotics by the subjects, but then no further statements are made relating to this. Considering that a wide array of drugs is used to “treat” FXS, why is there a mention of only this class of drugs?  Did the authors examine this further, and can any conclusion or statement be made as to the significance or relevance of antipsychotic use in the context of this study?

Response: We are grateful for the comment. Frequency of the antipsychotic use was assessed for to illustrate severity of neurobehavioral phenotype; indeed, all identified on it were males with FXS + ASD. Briefly, aripiprazole is preferred due to its favorable profile to address anxiety in FXS.

Lines 661-665 in Discussion: “As with ASD in males with severe ID, the relationship between FMRP levels and overall severe problem behavior (ABC-CFX total score ≥50) was also cofounded with greater cognitive impairment. These findings are in line with the recognition, in recent years, of a more severe neurobehavioral phenotype mainly in males with FXS. This is characterized by severe ID, ASD diagnosis, and severe irritability, aggression, agitation, and self-injury types of often anxiety-driven behavior [25, 27, 64]. Indeed, our sample of males with FXS and ASD [26, 34] had significantly elevated ABC-CFX total scores driven by the Irritability subscale [67], and higher proportion of anxiety diagnosis and CGI-SANX scores, respectively, which together required frequent use of atypical antipsychotic aripiprazole [6870].” 

  1. Why did the authors not examine or report how FMRP levels correlated with IQ or ID or some other behavioral parameter across all subjects and/or across subsets of subjects (i.e. correlation coefficients)?

Response: We are thankful for the reviewer’s comment.

As detailed in 2.5. Statistical analysis (Line 330), we focused our statistical analysis on Descriptive statistics and to compare means of two samples using both parametric and non-parametric test due to the limited power of our sample and subsamples, including permutation (five subjects). For example,

Lines 153-156. Detailed genomic profiling was performed for the subset of patients with FM (FMALL Subcohort, N = 37; Table 2); most of these participants also had FMRP measurements (FMFMRP Subcohort, N = 31; Table 1), detailed neurobehavioral assessments, and other genomic assays (Tables 2 & 3).

In Limitations (Lines 674-676), we also acknowledged that “Although our study is, to our knowledge, the largest molecular-neurobehavioral correlation using multiple and sensitive FMR1 and FMRP assays the sample size was relatively modest considering the variability and complexity of the molecular and phenotypical measures.”

Discussion

  1. “A significant finding from this study is that FMRP levels were two-fold higher in males with FXS and appreciable methylation mosaicism, expanding our knowledge on the link between methylation and FMR1 silencing.’ Two-fold higher compared to what, without MM?

Response: We are grateful for the comment. That is correct, the comparison was made with FXS males without MM.  This distinction has been incorporated into the revised submission (Line 617 and 618).

Lines 616-618A significant finding from this study is that FMRP levels were two-fold higher in males with FXS and appreciable methylation mosaicism compared to males without such mosaicism, expanding our knowledge on the link between methylation and FMR1 silencing.

  1. 11. “Availability of sensitive assays like qFMRP used here are key for advancing our understanding of the role of FMRP deficit in FXS, since threshold levels (<70%  typical) for ID have been recently described [19] and delineating differences among those with the lowest levels will require high power discrimination.” 

What does the term “threshold levels (<70% typical)” refer to – IQ, or FMRP?  If the latter, how is the threshold level determined or established?

Response: We are thankful for the comment. The threshold refers to IQ as an index of global intellectual disability. We clarified it in the text.

Line 669 in Discussion “Availability of sensitive assays like qFMRP used here are key for advancing our understanding of the role of FMRP deficit in FXS, since threshold levels (<70 % of typical IQ) for ID have been recently described [19] and delineating differences among those with the lowest levels will require high power discrimination [38, 53, 71].”

  1. The authors emphasize the utility of measuring FMRP. At the risk of opening a proverbial can of worms, could the authors add a few sentences summarizing (with references) what is known about the correlation between blood FMRP measurements and post mortem brain tissue FMRP levels in the same subject?

Response: We are grateful for the reviewer’s stimulating comment.

Lines 605-608 now incorporates it. “Moreover, very little is known about the correlation between blood FMRP measurements and post mortem brain tissue FMRP levels in the same subject [61, 62]. Only a single case study ‘verified’ reduced expression of FMR1 mRNA and FMRP in both peripheral blood and brain leading to the FXS [61].”

While this extensive review provide some really useful additional context, https://www.ncbi.nlm.nih.gov/pmc/articles/PMC5192497/ we acknowledge that it does not directly address the reviewer’s question.

  1. Minor comments

Typos “onsistent with 

 “….previous reports [39, 47, 52] and lyonization,”

In Table 5 it looks like a label for the first column is missing (N ?).

Response: We are thankful for the reviewer’s comment. We have carefully reviewed the text prior to resubmission. These typographic errors have been corrected.

Reviewer 2 Report

This work tries to pull together factors including size and methylation mosaicism as well as sex and to correlate these with IQ and behavioural phenotypes which is a significant piece of work.

Their findings include the effect of size mosaicism on FMRP expression and an inverse relationship between cognition and FMRP and differences in FMRP between the sexes - this is all in keeping with previous studies. They also note increasing FMRP with age which has also been reported.  They suggest in general ASD features are also linked to a lower level of FMRP, but not in the groups with lowest IQ.

I am unclear how the ASD diagnosis was defined - was this only on ADOS - or clinical impression - the number diagnosed on ADOS should be defined. There is a comment about longitudinal assessments (line 272) - I assume the phenotypic and behavioural data presented matches the time of sample collection

The authors use a control set to demonstrate a relationship between FMRP, FMR1 mRNA and methylation.  This holds well for a characterised control set and demonstrates reliability.  They do use lymphoblastoid cell lines in comparison to whole blood in the main study.  Is there any reason why the different sample types if one is a control dataset for the main study and is there data to confirm that the LCL cell lines are representative of fresh samples as per the main study.

There is a lot of discussion re methylation mosaicism (MM) but which samples does this include.  Methylation mosaicism is hard to determine from figure 2 as all samples appear to be >80% methylated - if methylation and or size mosaicism is present in some samples is there a more effective way to illustrate this. There also seems to be a broad range of FMRP levels even in the PM males - sample 24 seems to have less FMRP that some of the full mutation and fully methylated males - does this correlate with the phenotype in this male - can the authors comment. If we are to be believe the quantitation of FMRP is a reliable indicator of IQ and phenotype this need to be explained.

The authors indicate they measured mRNA in all subjects (or was this just the control group) - they do not comment on this in the discussion further. There has been some suggestion that elevated mRNA in FXS males may be linked to a higher ADOS severity score/ASD features.  Did the authors consider this? 

The authors note their ASD group is half the mean age of the non ASD group. As age is an noted variable for FMRP and is noted to increase with age does the relationship persist between FMRP and ASD if there is an age correction (or is there too small a cohort for such a correction).  Males with FXS when young and non verbal often seem to present with a more autistic phenotype. This should be explored in the text.

Author Response

This work tries to pull together factors including size and methylation mosaicism as well as sex and to correlate these with IQ and behavioural phenotypes which is a significant piece of work.

We appreciate and thank Reviewer 2.

Their findings include the effect of size mosaicism on FMRP expression and an inverse relationship between cognition and FMRP and differences in FMRP between the sexes - this is all in keeping with previous studies. They also note increasing FMRP with age which has also been reported.  They suggest in general ASD features are also linked to a lower level of FMRP, but not in the groups with lowest IQ.

Lines 640-647. We found two-fold lower levels of FMRP particularly in younger males (age- and IQ adjusted) with FXS and ASD than in those with FXS-only, although there was a wide range of protein expression in the latter group. Nonetheless, for the total FMFMRP male-only subset, this finding only held in ones with mild-moderate ID, as those with severe ID had even lower FMRP levels independent of ASD status. Since marked decrease in FMRP is not only the basis for FXS but also a contributor to the ASD phenotype, as the signaling pathways of FMR1 and other genes linked to the behavioral disorder substantially overlap [23, 24, 28, 63], a more definitive answer about the relationship between FMRP levels and ASD status deserves further study.

We are grateful for Reviewer 2 critiques, which we addressed below point-by-point.

  1. 1. I am unclear how the ASD diagnosis was defined - was this only on ADOS - or clinical impression - the number diagnosed on ADOS should be defined. There is a comment about longitudinal assessments (line 272) - I assume the phenotypic and behavioural data presented matches the time of sample collection.

Response:  We are thankful for the reviewer’s comment. We clarified the ambiguity as detailed below, including the longitudinal assessments termed here only to refer for clinically determining ASD diagnosis stability over time.

Lines 280-285 Diagnostic and Statistical Manual-5th Edition criteria (DSM-5) [51], supplemented by Autism Diagnostic Observation Schedule (ADOS) assessments available in males with FM, were used to diagnose ASD. All males with FM in the study diagnosed with ASD had both DSM-5 and ADOS assessments, respectively. The diagnosis of ASD (and Non-ASD) was made clinically, and confirmed longitudinally, for all subjects in the Clinical Cohort by a clinician (DBB) with expertise in idiopathic ASD, and ASD in FXS [25–27, 32, 34].

  1. The authors use a control set to demonstrate a relationship between FMRP, FMR1 mRNA and methylation.  This holds well for a characterized control set and demonstrates reliability.  They do use lymphoblastoid cell lines in comparison to whole blood in the main study.  Is there any reason why the different sample types if one is a control dataset for the main study and is there data to confirm that the LCL cell lines are representative of fresh samples as per the main study.

Response: We are thankful for the reviewer’s comment.

Additional support for the relevance and translation of cell line molecular analyses to blood specimens has been included through several text additions in Section 3.1 of the Results.  See in particular lines 353-355, 366-368 and 376-378 in the revised text.

Lines 353-355: “We note that the primary CGG sizing and AGG interruption genotypes in the immortalized cell lines were preserved when compared to results using these same assays for the original patient blood samples.”

Lines 366-368: “We also compared methylation levels of FM cell lines with the corresponding blood samples used for cell immortalization and found nearly identical percent methylation for all three sample pairs.”

Lines 376-378: “Further, the molecular profiles of each cell line were consistent with the corresponding clinical diagnoses and related presentations for the subjects that provided specimens to generate them (Fig. 1 and Table 4).”

  1. There is a lot of discussion re methylation mosaicism (MM) but which samples does this include.  Methylation mosaicism is hard to determine from figure 2 as all samples appear to be >80% methylated - if methylation and or size mosaicism is present in some samples is there a more effective way to illustrate this. There also seems to be a broad range of FMRP levels even in the PM males - sample 24 seems to have less FMRP that some of the full mutation and fully methylated males - does this correlate with the phenotype in this male - can the authors comment. If we are to be believe the quantitation of FMRP is a reliable indicator of IQ and phenotype this need to be explained.

Response: We are thankful for the reviewer’s comment.

Samples with and without methylation mosaicism are detailed in the Supplementary Materials.  This sample-by-sample accounting is provided for both blood and matched buccal specimens.  However, we failed to acknowledge the corresponding annotation in Fig. 2C, where the asterisk above the bars showing FMRP quantity marks samples without methylation mosaicism (as defined in the methods). 

We have corrected this omission in the revised submission. Figure 2 lines 420-421 “FM samples without methylation mosaicism are indicated with an asterisk above the bar.”

Indeed, the PM male- sample 24 has less FMRP than some males with the full mutation (FM) and fully methylated with mosaicism. We apologize that the above (now corrected) omission generated the confusion. Below, we have already detailed methylation mosaicism examples that were clinically less affected as a rule.

Lines 475-477Mosaicism, as we defined it, was (i) detected with high analytical sensitivity using assays previously known to report mosaicism down to 1-5% of cell equivalents [21, 57, 58], and (ii) detected in about 50% of the subjects in our cohort [Supplementary material]. 

To further illustrate, sample 42 is unmethylated FM male, which does correlate with the (less affected) phenotype in this male.

Lines 572-577 “To illustrate the potential clinical impact of the FMR1 and FMRP measurements reported here, one high functioning 7-year old male in the FXS-only group was an outlier with a normal FSIQ of 95 and an adaptive VABS-II Composite score of 79. His FMRP level was 7.4; (mean ± SD) all males (4.2 ± 3.3), FXS-only (5.4 ± 4.0). The subject had unmethylated size and methylation mosaicism on his blood sample, and a partially methylated PM smear with an additional PM peak on the buccal sample.”

In Limitation section (lines 676-681) we acknowledge that “Marked neurobehavioral differences between males and females with FXS, coupled with the well-known genetic differences inherent to X-linked disorders (e.g. X-inactivation in females), complicated the molecular and correlational analyses. Given the size of the cohort, the necessity of combining size- and methylation-mosaicism limited the full appreciation of differences between these types of mosaicism.”

  1. The authors indicate they measured mRNA in all subjects (or was this just the control group) - they do not comment on this in the discussion further. There has been some suggestion that elevated mRNA in FXS males may be linked to a higher ADOS severity score/ASD features.  Did the authors consider this? 

Lines 245-247. 2.3.4. FMR1 mRNA analysis

FMR1 transcript expression was evaluated via quantitative real-time PCR (qPCR) for all cell lines samples. We encountered that one FM cell line (RU08) was unmethylated (See below details in lines 358-362).

Lines 358-362. Two of the three FM cell lines were nearly completely methylated consistent with inactivation of FMR1 in FXS. However, one FM cell line (RU08) was unmethylated (Fig. 1C). Southern blot analysis of the original whole blood sample also showed lack of methylation (Table 4). To investigate this finding further, we isolated RNA and quantified the primary FMR1 mRNA isoform on all 11 cell lines.

  1. The authors note their ASD group is half the mean age of the non ASD group. As age is an noted variable for FMRP and is noted to increase with age does the relationship persist between FMRP and ASD if there is an age correction (or is there too small a cohort for such a correction).  Males with FXS when young and non-verbal often seem to present with a more autistic phenotype. This should be explored in the text.

Response: We are thankful for the reviewer’s another stimulating comment.

Indeed, our cohort was too small for the age correction. Nevertheless, our study did find that younger males below 11 years of age presented with ASD that also had lower FMRP values. We expanded these findings in Results and Discussion and Abstract as follows

Results Lines 566-571. Younger males (2-11 years, 58%) had two-fold lower FMRP levels (pg/ng) (N = 7/15, 3.2 ± 1.2 versus N = 8/15, 6.6 ± 4.4 FXS-only, p = 0.03, Welch’s t-test), adjusted for age-(6.9 ± 0.9 versus 6.9 ± 3.1 FXS-only, p = 0.49, Welch’s t-test) and FSIQ (44.1 ± 17.7 versus 59.6 ± 16.6 FXS-only, p = 0.12, Welch’s t-test); males with an ABC-CFX total score ≥50 and/or severe CGI-S scores (≥5), also closely linked to FXS + ASD, had significantly lower mean levels of FMRP (p = 0.01 and p = 0.05, respectively; Welch’s t-test).

Discussion Line 639-644. For instance, we found two-fold lower levels of FMRP particularly in young males (age- and IQ adjusted) with FXS and ASD (57% with severe ASD) than in those with FXS-only, although there was a wide range of protein expression in the latter group. Nonetheless, for the total FMFMRP male-only subset, this finding only held in ones with mild-moderate ID, as those with severe ID had even lower FMRP levels independent of ASD status.

Abstract lines 53-61 Molecular-neurobehavioral correlations confirmed the inverse relationship between overall severity of the FXS phenotype and decrease in FMRP levels (N = 26 males, mean 4.2 ± 3.3 pg/ng). Other intriguing findings included a relationship between the diagnosis of FXS with ASD and two-fold lower levels of FMRP (mean 3.21 ± 1.21 pg/ng, 57% with severe ASD) particularly in younger males (2-11 years), compared to FXS without ASD. Those with severe ID had even lower FMRP levels independent of ASD status in the male-only subset. The association between FMRP deficiency and overall severity of the neurobehavioral phenotype invites follow up studies in larger patient cohorts. They would be valuable to confirm and potentially extend our initial findings of the relationship between ASD and other neurobehavioral features and the magnitude of FMRP deficit.